# Sliding-Mode Control of Bidirectional Flyback Converters with Bus Voltage Regulation for Battery Interface

**Carlos Andres Ramos-Paja** [1,*] **, Juan David Bastidas-Rodriguez** [2] **and Luz Adriana Trejos-Grisales** [3]

1 Facultad de Minas, Universidad Nacional de Colombia, Medellín 050034, Colombia
2 Facultad de Ingeniería y Arquitectura, Universidad Nacional de Colombia, Manizales 170001, Colombia; jubastidasr@unal.edu.co
3 Facultad de Ingenierías, Instituto Tecnológico Metropolitano, Medellín 050028, Colombia; adrianatrejos@itm.edu.co
* Correspondence: caramosp@unal.edu.co

**Abstract:** Energy storage systems are essential for multiple applications like renewable energy systems, electric vehicles, microgrids, among others. Those systems are responsible of regulating the dc bus voltage using charging-discharging systems which are mainly formed by a power converter and a control system. This work focuses on the control system of a flyback converter. A detailed design procedure of an adaptive sliding-mode controller (SMC) and its parameters is presented. The proposed procedure was validated through simulations which allow to confirm its good performance in terms of global stability providing the desired dynamic of the dc bus voltage regulation.

**Keywords:** flyback; battery; sliding-mode; voltage regulation

## 1. Introduction

Energy storage systems (EESs) have experienced significant growth in the last years, which is evidenced in the 3.3 GWh and 3.1 GWh installed in 2018 and 2019, respectively [1]. Among the ESSs, Lithium-ion batteries play a key role since most of the new installed capacity corresponds to this technology [1]. Moreover, batteries are also essential in different applications like electric vehicles, microgrids, uninterruptible power supplies, stand-alone renewable energy systems, auxiliary services for the grid, among others. In most of those applications, the batteries compensate for the unbalances between generation and load by regulating the voltage of a dc bus through a charging-discharging system, which charges the batteries when generation power is greater than the load and discharges the battery in the opposite condition [2]. Moreover, the charging-discharging system holds the energy in the battery (i.e., idle mode) when the power and load are balanced [3].

A charging-discharging system is composed of two main elements, a bidirectional power converter and a control system [4]. On the one hand, the power converter allows the electrical connection between the battery and the dc bus because the battery voltage ($v_b$) is usually less than the dc bus voltage ($v_{bus}$). On the other hand, the control system generates the switching signals of the power converter to regulate $v_{bus}$ by the power exchange between the battery and the dc bus [5]. The dc bus voltage may be in the range of hundreds of volts [6], while the voltage of a single battery is usually 12 V [7]. Therefore, the battery pack in different applications are commonly formed by a set of batteries connected in series and parallel to reach voltage levels close to $v_{bus}$ and the required storage capacity, respectively. Moreover, the battery pack voltage should be close to $v_{bus}$ due to the limited voltage gain provided by the power converters commonly used in charging-discharging systems like Boost [8] and Buck-Boost [9]. The series connection of multiple batteries to form a battery pack may produce high temperatures and accelerated degradation in the individual batteries due to the differences in their electrical characteristics generated by non-uniform aging, small manufacturing defects, and the continuous cycles of charge and discharge [10]. One option to solve this problem is to use a balancing system, which

is an electronic system connected to a set of series-connected batteries that guarantees the uniformity in the state of charge of the individual batteries during the charging and discharging cycles [11]. Another option to solve the batteries unbalancing problem is to implement charging-discharging systems with power converters providing high-voltage gains, like a bidirectional flyback. In that way, each battery, or set of parallel-connected batteries, can be connected directly to the dc bus. This type of charging-discharging system eliminates the necessity of the series connection of batteries and balancing systems. Additionally, the storage capacity can be easily modified and the storage system may be formed by batteries with different technologies and characteristics as well as facilitating the system flexibility or the use of second-life batteries.

Flyback converter provides galvanic isolation and high voltage gain with a simple hardware structure, which are characteristics not provided by the converters typically used. Hence, a flyback converter is an interesting option to implement a battery charging-discharging system [12,13]. In fact, this converter has been used in battery charging-discharging system to regulate a dc bus voltage [12,13], in battery charging systems for different applications [14], as well as in other applications that utilize batteries [15]. In the literature, it is possible to find different control strategies and design procedures for flyback converters used in battery charging applications. Linear controllers are the most widely used, particularly PI [13,15,16] and two-poles two-zeros [17,18] compensators; nonetheless, other authors prefer a sliding mode controller (SMC) [12], a peak current control [19] or an open-loop controller [14]. Moreover, in many cases, the controller is not reported or explained [20,21]. In [16,22] the authors propose a battery charger for an electric vehicle (EV) formed by two converters, the first one is a Landsman converter [16,23] or a Cuk converter [22] feed by the grid, and the second one is a flyback converter connected to the EV battery. These papers implement a cascade controller for the flyback converter, where the inner and outer loops regulate the battery current and voltage, respectively. The compensators in both loops are PIs whose integral and proportional constants can be tuned to obtain the desired phase margin and crossover frequency as proposed in [16]. Nevertheless, other papers do not propose a design procedure or do not provide information on the PIs parameters [22].

In the literature, there are also reported other applications that implement a flyback converter to charge [24,25] or discharge [15,17] a battery, which only use one linear compensator to regulate the flyback output voltage. For example, a flyback converter is used in [24] and [25] to charge the battery of a phone and as an auxiliary battery charger in a microgrid, respectively. In both cases, a PI is implemented to regulate the battery voltage during the constant-voltage charging; while an open-loop controller is implemented during the constant-current charging. The PI proposed in [24] is tuned by evaluating three values for the integral constant to obtain a fast response; then, three values of the proportional constant are evaluated to eliminate the overshoot. However, in [25] the authors do not provide any design procedure or guideline to design the parameters of the PI regulator. Additionally, in [15] a flyback converter is used to regulate the voltage of a LED luminary by discharging a battery where a PI is used to regulate the luminary voltage; nonetheless, the paper does not include the design procedure of the PI parameters. On the other hand, in [17,18] two-poles two-zeros compensators are utilized to regulate the voltage of a dc motor and a brushless dc motor supplied by a battery. Both papers use a flyback converter and the proposed compensators are tuned by using the frequency response; however, only the authors of [17] define the desired frequency response parameters (i.e., gain margin and crossover frequency).

Peak current controllers have also been applied to control flyback converters in battery and supercapacitor charging applications, as reported in [19,26]. In both papers a flyback converter is used to implement an ESS charger with power factor correction; however, the proposed controllers are different. On the one hand, the peak current controller proposed in [19] includes an exponential compensation ramp to generate a rectified sinusoidal current at the input of the flyback converter (i.e., output of the diode bridge rectifier). Such a compensation ramp has two parameters whose design procedure is included in the paper. On the other hand, the peak current controller proposed in [26]

uses the flyback output and input voltages as well as the MOSFET current to determine the converter duty cycle; then, the flyback PWM is generated with a 555 integrated circuit. Nevertheless, both papers do not include any stability analysis of the proposed controller and it is not clear how the battery voltage and/or current are regulated. From the papers discussed before it can be observed that most of the reported controllers are linear and conceived for regulating the battery voltage and/or current during the battery charging or discharging; hence, they cannot guarantee the system stability for any operating condition and for the different battery operating modes (i.e., charging, discharging, and idle).

Moreover, the flyback-based battery charging-discharging systems proposed in [12,13] are aimed at regulating the dc bus voltage of a dc microgrid. On the one hand, in [13] the authors propose a cascade linear controller, where the inner loop is an adaptive proportional compensator of the magnetizing current ($i_m$) and the outer loop is an adaptive PI regulator of the dc bus voltage ($v_{bus}$). This controller can be applied for and exiting converter and its design procedure considers the converter's parameters; however, the cascade structure limits the dynamic response of the system since the crossover frequency of the inner and outer loops are $1/5$ and $1/25$ of the switching frequency, respectively. On the other hand, in [12] the authors propose an adaptive SMC where the sliding surface includes the dc bus voltage error, $i_m$, and the dc bus current ($i_{bus}$) that guarantees global stability. The paper also includes a co-design procedure of the SMC and the converter parameters; nevertheless, it cannot be applied to an existing converter, which limits its application to new converters. Additionally, both controllers require the measurement of $i_{bus}$, which is difficult to implement in a microgrid since this current is the algebraic sum of all the currents drained from or supplied to the dc bus.

This paper proposes an adaptive SMC for a dc bus voltage regulation based on a flyback converter along with a detailed design procedure of the SMC's adaptive and constant parameters. The SMC's switching function uses the dc bus voltage error, its integral, and $i_m$, but it doesn't require the measurement of $i_{bus}$ nor the design of the power converter. Moreover, the proposed design procedure guarantees the global stability of the system and considers the characteristics of the flyback converter, the desired dynamic performance of dc bus voltage regulation, and the maximum switching frequency of the flyback MOSFETs. The paper begins with the electrical (Section 2) and mathematical (Section 3) models of the dc bus voltage regulation system. Then, it follows with the analysis of the proposed SMC including the sliding surface design, the stability analysis, and the description of the closed-loop dynamics of the controlled system in Section 4. Later, the SMC's design procedure is introduced in Section 5 and the practical implementation of the proposed controller is discussed in Section 6. The proposed design procedure is validated in Section 7 with different simulations, which show that the proposed adaptive SMC guarantees global stability in different operating conditions and the three operating modes (i.e., charging, discharging, and idle) even with step perturbations in the dc bus current. Finally, the conclusions close the paper in Section 8.

## 2. Circuital Model

The circuital implementation of the battery interface, based on a flyback converter, is depicted in Figure 1. Such a circuital representation models the flyback transformer using both magnetizing ($L_m$) and leakage ($L_k$) inductances, where the transformer has a general turn-ratio $1 : n$. The main objectives of this power system are to provide both power balance and regulated voltage to the dc bus, thus the flyback converter must be based on a bidirectional topology:

- Discharge: in this operation state the current flows from the battery to the dc bus, thus the switch $S_1$ works as the MOSFET of the typical flyback topology, controlled by the main control signal u; while switch $S_2$ works as the diode of the flyback topology, hence it is activated with a complementary signal $\bar{u} = 1 - u$. In this operation, the currents of both the battery ($i_b$) and dc bus ($i_{bus}$) are considered positive since power is delivered to the bus.

- Charge: in this operation state the current flows from the dc bus to the battery, hence the switch $S_1$ works as the diode of the typical flyback topology, and switch $S_2$ works as the MOSFET of the flyback topology. In this operation, the currents of both the battery and dc bus are considered negative since power is extracted from the bus.
- Idle: in this operation state no power flows between the dc bus and the battery, thus the average values of both battery and dc bus currents are equal to zero.

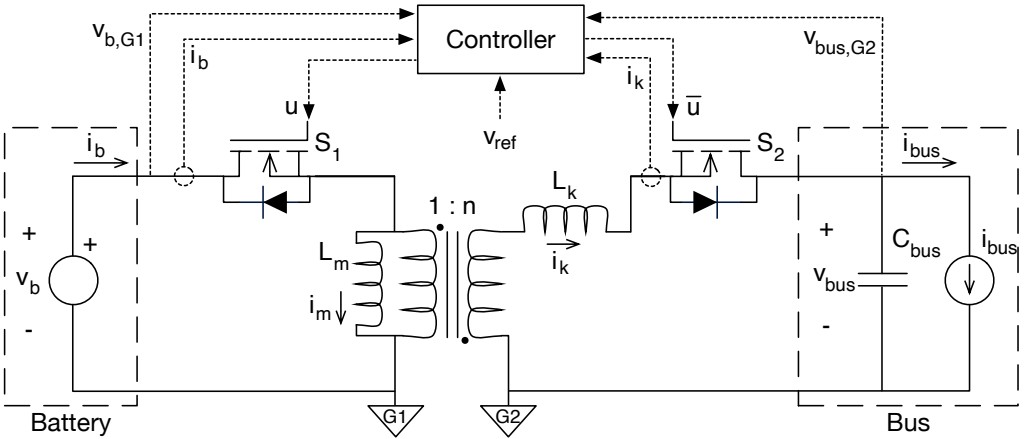

**Figure 1.** Circuital implementation of the battery interface.

In this circuital representation, the dc bus is modeled by a capacitance ($C_{bus}$) and a current source ($i_{bus}$). The capacitance is the result of the parallel connection of the capacitors used in the filter stages of all the devices connected to the bus; while the current source represents the current needed to ensure the power balance in the bus, i.e., discharge state if the sources produce lower power than the loads' consumption, charge state if the sources produce higher power than the loads' consumption, and idle state if the sources produce the same power consumed by the loads.

The flyback transformer of the circuit provides galvanic isolation between the battery and the dc bus, which is useful to protect the battery from failures in the dc bus. Such galvanic isolation forces the definition of two voltage grounds, one for the battery side (named G1) and another for the bus side (named G2). Therefore, the battery voltage $v_b$ is measured with respect to G1, while the bus voltage $v_{bus}$ is measured with respect to G2; those conditions are represented in the circuit of Figure 1 with the corresponding sub-indexes for the measured signals (dotted lines $v_{b,G1}$ and $v_{bus,G2}$). The circuit also shows the measurements used to process the system control: battery voltage $v_{b,G1}$, battery current $i_b$, bus voltage $v_{dc,G2}$, and leakage current $i_k$, where $i_k$ is measured at the secondary side of the transformer. Finally, the controller receives the reference value $v_{ref}$ for the bus voltage, and the outputs are both main u and complementary $\bar{u}$ control signals.

## 3. Mathematical Model

The previous power system must be modeled to design the sliding-mode controller. Such a modeling process is supported using the circuital topologies depicted in Figure 2, which are the result of settling the control signal to $u = 1$ and $u = 0$, respectively.

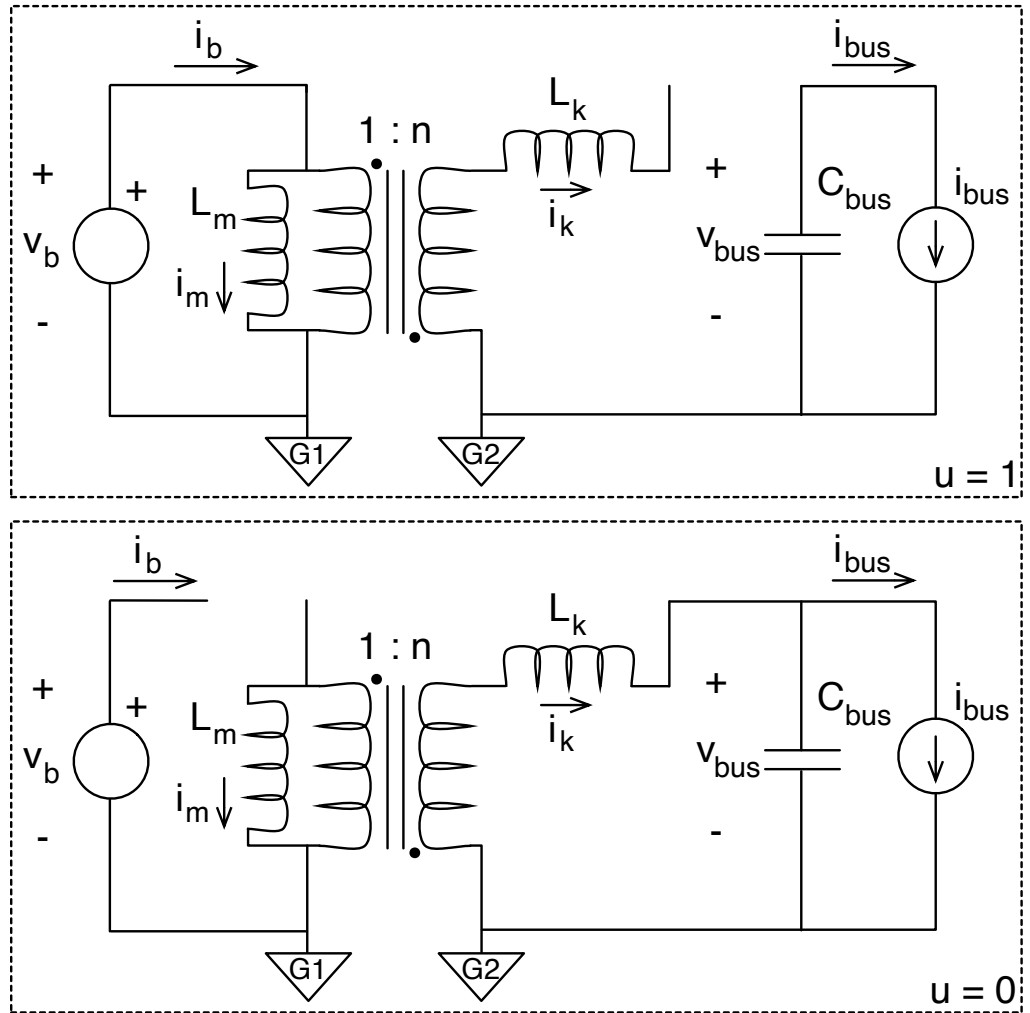

**Figure 2.** Circuital topologies of the battery interface.

For the first topology, switch $S_1$ is closed ($u = 1$) and switch $S_2$ is open ($\bar{u} = 0$), forming the circuit at the top of Figure 2. In such a circuit, the equations describing the magnetizing current $i_m$, bus voltage $v_{bus}$, and leakage current $i_k$ are:

$$\frac{di_m}{dt} = \frac{v_b}{L_m} \tag{1}$$

$$\frac{dv_{bus}}{dt} = \frac{-i_{bus}}{C_{bus}} \tag{2}$$

$$i_k = 0 \tag{3}$$

For the second topology, switch $S_1$ is open ($u = 0$) and switch $S_2$ is closed ($\bar{u} = 1$), forming the circuit at the bottom of Figure 2. The equations describing the main electrical variables are:

$$\frac{di_m}{dt} = \frac{-v_{bus}}{n \cdot L_m + L_k/n} \tag{4}$$

$$\frac{dv_{bus}}{dt} = \frac{-i_{bus} + i_m/n}{C_{bus}} \tag{5}$$

$$i_k = i_m/n \tag{6}$$

Combining the previous topological equations, using the control signal $u$, leads to the switched model of the battery interface, as follows:

$$\frac{di_m}{dt} = \frac{v_b \cdot u}{L_m} - \frac{v_{bus} \cdot (1-u)}{n \cdot L_m + L_k/n} \tag{7}$$

$$\frac{dv_{bus}}{dt} = \frac{(i_m/n) \cdot (1-u) - i_{bus}}{C_{bus}} \tag{8}$$

$$i_k = (i_m/n) \cdot (1-u) \tag{9}$$

The previous model provides a precise description of the converter operation, since the binary change of the control signal $u$ enables the reproduction of the switching ripple in the electrical variables. However, when the converter is driven by a modulator, i.e., a PWM circuit or a sliding-mode controller, the control signal $u$ is generated by the modulator and the converter operation could be described in terms of the duty-cycle $d$. Taking into account that the duty cycle is the average value of the control signal inside the switching period $T_{sw}$, as given in (10), an averaged model can be designed by averaging the switched differential Equations (7)–(9) within $T_{sw}$, in which the independent variable is the duty cycle. Such an averaged model is reported in (11)–(13).

$$d = \frac{1}{T_{sw}} \cdot \int_0^{T_{sw}} u \, dt \tag{10}$$

$$\frac{di_m}{dt} = \frac{v_b \cdot d}{L_m} - \frac{v_{bus} \cdot (1-d)}{n \cdot L_m + L_k/n} \tag{11}$$

$$\frac{dv_{bus}}{dt} = \frac{(i_m/n) \cdot (1-d) - i_{bus}}{C_{bus}} \tag{12}$$

$$i_k = (i_m/n) \cdot (1-d) \tag{13}$$

The stable values of $i_m$ and the duty cycle $d$ are obtained by assuming the derivatives of (11) and (12) are equal to zero:

$$d = \frac{v_{bus}}{v_{bus} + v_b \cdot \left(n + \frac{L_k}{n \cdot L_m}\right)} \tag{14}$$

$$i_m = n \cdot i_{bus} \cdot \frac{v_{bus}}{v_b} \cdot \left(1 + \frac{1}{n + \frac{L_k}{n \cdot L_m}}\right) \tag{15}$$

Finally, the ripples in both the magnetizing current ($r_{i_m}$) and bus voltage ($r_{v_{bus}}$) are calculated from (1), (2), and (10) as follows:

$$r_{i_m} = \frac{v_b \cdot d \cdot T_{sw}}{2 \cdot L_m} \tag{16}$$

$$r_{v_{bus}} = \frac{i_{bus} \cdot d \cdot T_{sw}}{2 \cdot C_{bus}} \tag{17}$$

## 4. Design of the Sliding-Mode Controller

The design of the sliding-mode controller (SMC) is based on a suitable sliding surface, which defines the behavior of the system under control. After designing the sliding-surface, the feasibility of such a surface must be tested using the transversality criteria, which evaluates the ability of the SMC to modify the behavior of the power system, thus executing a control law. Two additional tests are needed to ensure the global stability: the reachability conditions, which describe the conditions needed to ensure the system is always driven to enter into the desired sliding surface; and the equivalent control condition, which describes the conditions needed to keep trapped the system into the desired sliding surface. Finally, the closed-loop behavior of the system must be modeled to provide design guidelines for the surface parameters. Those procedures are described in the following subsections.

### 4.1. Sliding Surface Design

The design of the sliding surface will be performed by analyzing the bus node in Figure 1, where the averaged value of the leakage current, given in (13), flows through the switch S$_2$. Thus, it is evident that the averaged value of the $i_k$ current is the only controlled variable able to regulate the bus voltage. However, since $i_k = 0$ when $u = 1$ as reported in (3), then $i_k$ is a discontinuous variable, thus it must not be included in the sliding surface to avoid a discontinuous surface. Instead, the averaged $i_k$ value is expressed in (13) as a function of the magnetizing current $i_m$, which is a continuous variable. Therefore, the magnetizing current is included in the sliding surface. Taking into account that the main objective of the battery interface is to provide voltage regulation to the dc bus, the error between the desired bus voltage $v_{ref}$ and the actual bus voltage $v_{bus}$ must be inserted into the sliding surface. Finally, the integral of the voltage error is also introduced to avoid steady-state errors on the bus voltage. The resulting sliding surface $S_{(X)}$ is reported in (18), where the switching function $X$ of the proposed sliding surface is given in (19), where $a$ and $b$ are the parameters of the SMC.

$$S_{(X)} = \{X = 0\} \tag{18}$$

$$X = i_m + a \cdot \left(v_{bus} - v_{ref}\right) + b \cdot \int \left(v_{bus} - v_{ref}\right) dt \tag{19}$$

The time derivative of the switching function $X$ is needed to perform the stability analysis; hence, it is obtained by deriving Equation (19) and replacing expressions (7) and (8), as given in (20). Such an expression considers the reference value $v_{ref}$ constant, which is the most common practice to provide a save and regulated bus voltage, thus the time derivative of $v_{ref}$ is zero, i.e., $dv_{ref}/dt = 0$.

$$\frac{dX}{dt} = \frac{v_b \cdot u}{L_m} - \frac{v_{bus} \cdot (1 - u)}{n \cdot L_m + L_k/n} + a \cdot \frac{(i_m/n) \cdot (1 - u) - i_{bus}}{C_{bus}} + b \cdot \left(v_{bus} - v_{ref}\right) \tag{20}$$

### 4.2. Stability Restrictions

The first condition to test is the ability of the controller to modify the behavior of the power converter, which is named as transversality condition [27]: to define the system trajectory it is needed to modify the value of the switching function, thus the control signal must be present into the derivative of the switching function. This restriction is formalized as follows:

$$\frac{d}{du}\left(\frac{dX}{dt}\right) \neq 0 \tag{21}$$

Then, the transversality condition is tested by replacing expression (20) into (21):

$$\frac{d}{du}\left(\frac{dX}{dt}\right) = \frac{v_b}{L_m} + \frac{v_{bus}}{n \cdot L_m + L_k/n} - \frac{a \cdot i_m}{n \cdot C_{bus}} \tag{22}$$

Therefore, to ensure the SMC's ability to modify the system behavior, Equation (22) must be different from zero, thus either positive or negative. However, the analysis of the reachability conditions require a defined sign of the transversality condition; therefore, expression (22) must be always positive or always negative. Considering that expression (22) is positive for $i_m = 0$, which is a possible value for $i_m$ under the idle state, it is evident that expression (22) must be always positive. The magnetizing current $i_m$ could be positive (battery discharge), negative (battery charger), or zero (idle). In the idle case expression (22) is positive since $v_b$, $L_m$, $L_k$, $n$, $v_{bus}$, and $C_{bus}$ are positive values. Moreover, $a$ and $b$ must be selected as positive values to ensure stability, as will be demonstrated in Section 4.3; thus, a negative value of $i_m$ ensures a positive value for expression (22). However, a positive value of $i_m$ could produce a negative value of expression (22); hence, the battery discharge state ($i_m > 0$) is the most restrictive condition. Finally, the limit value of $a$ to ensure positive

transversality is obtained by assuming expression (22) positive and $i_m > 0$, which is the first stability restriction:

$$a < \frac{C_{bus}}{i_{bus} \cdot \frac{v_{bus}}{v_b} \cdot \left( \frac{L_m}{n \cdot L_m + L_k/n} + 1 \right)} \cdot \left[ \frac{v_b}{L_m} + \frac{v_{bus}}{n \cdot L_m + L_k/n} \right] \tag{23}$$

The next stability restrictions are obtained from the reachability conditions:

- Operating under the sliding surface ($X < 0$) requires a positive derivative ($dX/dt > 0$) to reach the surface.
- Operating above the sliding surface ($X > 0$) requires a negative derivative ($dX/dt < 0$) to reach the surface.

The previous conditions are formalized as follows:

$$\lim_{X \to 0-} \frac{dX}{dt} > 0 \quad \wedge \quad \lim_{X \to 0+} \frac{dX}{dt} < 0 \tag{24}$$

However, the sign of the switching function derivative depends on the sign of the transversality; a positive transversality value $\frac{d}{du}\left(\frac{dX}{dt}\right) > 0$ imply that positive changes on $u$ ($0 \to 1$) produce a positive switching function derivative $\frac{dX}{dt} > 0$, while negative changes on $u$ ($1 \to 0$) produces $\frac{dX}{dt} < 0$. Under the light of the previous analyses, the restrictions given in (24) are rewritten as follows:

$$\lim_{X \to 0-} \frac{dX}{dt}\bigg|_{u=1} > 0 \tag{25}$$

$$\lim_{X \to 0+} \frac{dX}{dt}\bigg|_{u=0} < 0 \tag{26}$$

The maximum safe deviation of the bus voltage is defined as $e_{bus} = \max(v_{bus}) - v_{ref}$, and evaluating (25) using expression (20), leads to the second stability restriction reported in (27), which must be evaluated for the maximum values of $i_{bus}$ and $e_{bus}$, both positive and negative.

$$\frac{v_b}{L_m} - \frac{a \cdot i_{bus}}{C_{bus}} + b \cdot e_{bus} > 0 \tag{27}$$

Similarly, evaluating (26) using expression (20) leads to the third stability restriction reported in (28), which also must be evaluated for the maximum values of $i_{bus}$ and $e_{bus}$, both positive and negative.

$$-1 + \frac{a \cdot i_{bus} \cdot L_m}{v_b \cdot C_{bus}} + \frac{b \cdot e_{bus} \cdot (n \cdot L_m + L_k/n)}{v_{bus}} < 0 \tag{28}$$

Finally, the equivalent control condition verifies that the average value of the control signal, i.e., the duty cycle, is always inside the valid range $(0,1)$. Therefore, this condition is used to ensure that the duty cycle is never saturated. The evaluation of the equivalent control assumes the operation within the sliding surface, thus $X = 0$ and $\frac{dX}{dt} = 0$. Then, the average value of $u$ is calculated from $\frac{dX}{dt} = 0$, replacing expression (20), as follows:

$$0 < d = \frac{\frac{v_{bus}}{n \cdot L_m + L_k/n} - \frac{a \cdot i_m}{C_{bus} \cdot n} + \frac{a \cdot i_{bus}}{C_{bus}} - b \cdot e_{bus}}{\frac{v_b}{L_m} + \frac{v_{bus}}{n \cdot L_m + L_k/n} - \frac{a \cdot i_m}{C_{bus} \cdot n}} < 1 \tag{29}$$

Evaluating expression (29) leads to the same stability restriction previously reported in (27), and (28). Therefore, fulfilling the stability restrictions (23), (27), and (28) ensures the SMC's ability to reach the sliding surface and keep trapped inside without saturating the duty cycle, which ensures global stability. In conclusion, those stability restrictions must be

considered for the calculation of the surface parameters $a$ and $b$, which will be discussed in Section 5.

*4.3. Closed-Loop Dynamics*

The next step is to model the behavior of the system under the action of the SMC. This analysis is performed using a simplified model of the bus node given in Figure 1. From the averaged model given in (13) is noted that the average value of $i_k$ is $(i_m/n) \cdot (1-d)$, and the global stability of the SMC ensures that $i_m = -a \cdot \left(v_{bus} - v_{ref}\right) - b \cdot \int \left(v_{bus} - v_{ref}\right) dt$ as given in (18) and (19). Therefore, the closed loop dynamic behavior of the bus voltage is described by the following differential equation:

$$\frac{dv_{bus}}{dt} = \frac{\left(\frac{1-d}{n}\right) \cdot i_m - i_{bus}}{C_{bus}} \tag{30}$$

Applying the Laplace transformation:

$$V_{bus}(s) = \frac{\frac{a \cdot (1-d)}{C_{bus} \cdot n} \cdot s + \frac{b \cdot (1-d)}{C_{bus} \cdot n}}{s^2 + \frac{a \cdot (1-d)}{C_{bus} \cdot n} \cdot s + \frac{b \cdot (1-d)}{C_{bus} \cdot n}} \cdot V_{ref}(s) - \frac{\frac{s}{C_{bus}}}{s^2 + \frac{a \cdot (1-d)}{C_{bus} \cdot n} \cdot s + \frac{b \cdot (1-d)}{C_{bus} \cdot n}} \cdot I_{bus}(s) \tag{31}$$

The previous expression describes the dynamic behavior of the bus voltage to changes on both the reference value $V_{ref}(s)$ and the bus current $I_{bus}(s)$. However, as previously discussed in Subsection 4.1, the reference value is considered constant to provide a save and regulated bus voltage; thus, the transfer function $G_{bus}$ describing the changes on the bus voltage caused by bus current perturbations is given below:

$$G_{bus}(s) = \frac{V_{bus}(s)}{I_{bus}(s)} = \frac{-\frac{s}{C_{bus}}}{s^2 + \frac{a \cdot (1-d)}{C_{bus} \cdot n} \cdot s + \frac{b \cdot (1-d)}{C_{bus} \cdot n}} \tag{32}$$

In the previous equivalent transfer function $G_{bus}$ the capacitance $C_{bus}$, duty cycle $d$ and turn ratio $n$ are positive values, thus applying the Routh-Hurwitz theorem [28] to the denominator of (32) confirms that both $a$ and $b$ must be positive values to ensure stable (negative) equivalent poles. Finally, the previous transfer function is used to design both $a$ and $b$ in agreement with the performance criteria needed for the bus voltage; that process will be described in the following section.

## 5. Calculation of the SMC Parameters

The main problem of using $G_{bus}$ to design the bus voltage behavior concerns the changes in the transfer function caused by perturbations on the duty cycle $d$, which occur when the bus voltage is perturbed due to variations on the bus current. Therefore, transfer function $G_{bus}(s)$ must be normalized in terms of the duty cycle using the transformation given in (33), where $k$ adapts the new parameters $\alpha$ and $\beta$ to include changes on the duty cycle.

$$k = \frac{n}{1-d} \quad \Rightarrow \quad \left\{ \alpha = \frac{a}{k} \quad \wedge \quad \beta = \frac{b}{k} \right\} \tag{33}$$

Therefore, the parameters $\alpha$ and $\beta$ are normalized with respect to the duty cycle, which produces the normalized transfer function $\Gamma_{bus}$:

$$\Gamma_{bus}(s) = \frac{V_{bus}(s)}{I_{bus}(s)} = \frac{-\frac{s}{C_{bus}}}{s^2 + \frac{\alpha}{C_{bus}} \cdot s + \frac{\beta}{C_{bus}}} \tag{34}$$

The poles $\sigma_{1,2}$ of the normalized transfer function $\Gamma_{bus}(s)$ are given in (36), where the restriction $\alpha > 2 \cdot \sqrt{\beta \cdot C_{bus}}$ is imposed to avoid complex poles. Such a restriction ensures

an over-damped response of the bus voltage, thus no voltage oscillations occur after a bus current perturbation takes place.

$$\sigma_{1,2} = \frac{-\frac{\alpha}{C_{bus}} \pm \sqrt{\left(\frac{\alpha}{C_{bus}}\right)^2 - 4 \cdot \frac{\beta}{C_{bus}}}}{2} \qquad (35)$$
$$\text{with} \quad \alpha > 2 \cdot \sqrt{\beta \cdot C_{bus}}$$

The behavior of the bus voltage can be defined by analyzing the response to bus current perturbations; the worst case corresponds to a step current perturbation $I_{bus}(s) = I_{bus}/s$, where $I_{bus}$ is the magnitude of the step current. Then, the Laplace response of the normalized transfer function (34) to the step current is given in (36), where $\sigma_2$ and $\sigma_1$ are the poles previously reported in (36).

$$V_{bus}(s) = \frac{I_{bus}}{C_{bus} \cdot (\sigma_2 - \sigma_1)} \cdot \left[ \frac{1}{s - \sigma_1} - \frac{1}{s - \sigma_2} \right] \qquad (36)$$

To design the bus voltage response in terms of time-domain criteria, the Laplace function (36) must be transformed to a time-domain waveform using the inverse Laplace transformation, as follows:

$$v_{bus}(t) = \frac{I_{bus} \cdot \left(e^{\sigma_1 \cdot t} - e^{\sigma_2 \cdot t}\right)}{C_{bus} \cdot (\sigma_2 - \sigma_1)} \qquad (37)$$

The bus voltage response is designed using the following criteria:

- Voltage deviation ($\Delta V_{bus}$): the maximum bus voltage deviation caused by a step current perturbation with magnitude $I_{bus}$.
- Settling time ($t_s$): the maximum time acceptable to compensate the voltage deviation and enter into a safe voltage band $[-\epsilon, \epsilon]$ around the reference value $v_{ref}$.

The previous criteria are illustrated in Figure 3, which shows the bus voltage response after a positive step current occurs in the bus ($I_{bus}$).

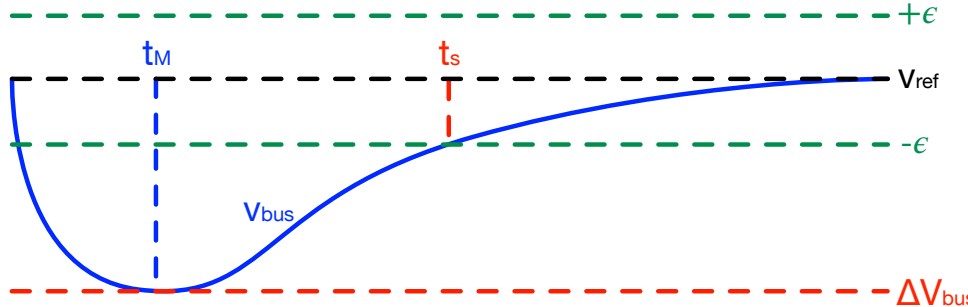

**Figure 3.** Design criteria of the bus voltage behavior ($\Delta V_{bus}$ and $t_s$).

The first criterion ($\Delta V_{bus}$) is calculated when the time-derivative of Equation (37) is equal to zero, thus $\frac{dv_{bus}}{dt} = 0$, which occurs when $t = t_M$:

$$t_M = \frac{\ln\left(\frac{\sigma_1}{\sigma_2}\right)}{\sigma_2 - \sigma_1} \qquad (38)$$

Therefore, the maximum bus voltage deviation is obtained by evaluating the time-domain waveform (37) at $t_M$:

$$\Delta V_{bus} = \frac{I_{bus} \cdot \left[ e^{\frac{\sigma_1}{\sigma_2 - \sigma_1} \cdot \ln\left(\frac{\sigma_1}{\sigma_2}\right)} - e^{\frac{\sigma_2}{\sigma_2 - \sigma_1} \cdot \ln\left(\frac{\sigma_1}{\sigma_2}\right)} \right]}{C_{bus} \cdot (\sigma_2 - \sigma_1)} \tag{39}$$

The second criterion $(t_s)$ is calculated when $v_{bus} = \epsilon \cdot v_{ref}$, which occurs after the maximum bus voltage deviation has been compensated, thus $t_s > t_M$ as it is depicted in Figure 3. The resulting equation needed to calculate $t_s$, obtained from (37), is the following one:

$$e^{\sigma_1 \cdot t_s} - e^{\sigma_2 \cdot t_s} = \frac{C_{bus} \cdot \left(\epsilon \cdot v_{ref}\right) \cdot (\sigma_2 - \sigma_1)}{I_{bus}} \tag{40}$$
$$\text{with} \quad t_s > t_M$$

Then, poles $\sigma_1$ and $\sigma_2$ are calculated from the non-linear equation system formed by (39) and (41), ensuring that both $\Delta V_{bus}$ and $t_s$ restrictions are fulfilled. Those normalized poles are used to calculate the normalized parameters $\alpha$ and $\beta$, from (36), as follows:

$$\alpha = C_{bus} \cdot (\sigma_1 + \sigma_2) \quad \wedge \quad \beta = C_{bus} \cdot \sigma_1 \cdot \sigma_2 \tag{41}$$

The $\alpha$ and $\beta$ parameters are calculated offline to impose the desired bus voltage behavior, but the switching function (19) of the sliding surface is described in terms of the non-normalized parameters $a$ and $b$, thus those parameters must be calculated, in real-time, to compensate the changes on the duty cycle:

$$a = \alpha \cdot k \quad \wedge \quad b = \beta \cdot k \quad \text{with} \quad k = \frac{n}{1 - d} \tag{42}$$

Finally, the parameters $a$ and $b$ are dynamically adapted, using $k$, to implement an adaptive SMC providing the same dynamic performance for all the operation conditions: charge, discharge, and idle. The practical implementation of the proposed SMC is discussed in the following section.

## 6. Practical Implementation of the SMC

Theoretical sliding mode controllers consider an infinite switching frequency around the steady-state conditions, which is impossible for a practical implementation [29]. Therefore, practical implementations require relaxing the sliding surface by introducing a hysteresis band $[-H, +H]$ around the sliding surface to limit the switching frequency, as follows:

$$S_{(X)} = \{-H < X < +H\} \tag{43}$$

The previous practical implementation of the sliding surface considers the same switching function $X$ given in (19), thus all the stability analyses and design equations hold. The hysteresis band expressed in (43) requires the SMC to act when $X$ reaches the hysteresis limits, as follows:

- When $X = +H$, the switching function derivative must be set to a negative value, which prevents $X$ to become higher than $+H$ ($X > +H$). The reachability condition given in (26) shows that such a negative switching function derivative requires setting the control signal to $u = 0$.
- When $X = -H$, the switching function derivative must be set to a positive value, which prevents $X$ to become lower than $-H$ ($X < -H$). The reachability condition given in (25) shows that such a positive switching function derivative requires setting the control signal to $u = 1$.

The previous behavior is formalized using the boolean control law reported in (44), which is used to implement the SMC. Such a control law can be easily implemented using two comparators and a S-R flip-flop as reported in the switching circuit of Figure 4, where the main parameter is the hysteresis width $H$.

$$\left\{ \begin{array}{lcl} X >= +H & \rightarrow & u = 0 \\ X <= -H & \rightarrow & u = 1 \end{array} \right\} \tag{44}$$

Since the main objective of the hysteresis band is to limit the switching frequency $F_{sw}$, the parameter $H$ must be described in terms of $F_{sw}$. Such a switching frequency is the result of the switching action of the MOSFETs [30], which also produces the switching ripple on both the magnetizing current $r_{i_m}$ and bus voltage $r_{v_{bus}}$ previously reported in (16) and (17), respectively. Taking into account the switching function $X$ depends on both $i_m$ and $v_{bus}$ (19), the ripple of $X$ is calculated as given in (45) because the charge balance principle ensures that the integral of the voltage ripple $r_{v_{bus}} = v_{bus} - v_{ref}$ is equal to zero in steady-state.

$$r_X = r_{i_m} + a \cdot r_{v_{bus}} \tag{45}$$

Replacing the expressions of (16) and (17) into (45), considering the duty cycle expression of (14) and the equivalence $T_{sw} = 1/F_{sw}$, results in the value of $H = r_X$ ensuring a maximum steady-state switching frequency $F_{sw}$ calculated in (46). Such an expression reports that the higher switching frequency occurs in the charge state ($i_{bus} < 0$), while the discharge state ($i_{bus} > 0$) exhibits a lower switching frequency. Therefore, the parameter $H$ must be evaluated for the most negative bus current $i_{bus} = -I_{bus}$, which corresponds to the higher charging current.

$$H \geq \frac{1}{2 \cdot F_{sw}} \cdot \left[ \frac{v_b}{L_m} - \frac{a \cdot i_{bus}}{C_{bus}} \right] \cdot \left[ \frac{v_{bus}}{v_{bus} + v_b \cdot \left( n + \frac{L_k}{n \cdot L_m} \right)} \right] \tag{46}$$

$$\text{with} \quad i_{bus} = -I_{bus}$$

The final step for the SMC implementation is to calculate, in real-time, the switching function $X$. However, the calculation of $X$ requires the value of $i_m$, which cannot be measured in a single point. Instead, from the topologies in Figure 2 it is observed that $i_m = i_b$ when $u = 1$ and $i_m = n \cdot i_k$ when $u = 0$, thus $i_m = i_b$ is used to evaluate the upper limit of the hysteresis band ($+H$), while $i_m = n \cdot i_k$ is used to evaluate the lower limit of the band ($-H$). Figure 4 shows the practical implementation of the proposed SMC, which requires the measurement of the battery voltage ($v_b$), bus voltage ($v_{bus}$), battery current ($i_b$), and leakage current ($i_k$). Such an implementation requires adders, subtractors, gains, integrals, multipliers, and dividers, which can be implemented using operational amplifiers and other integrated circuits. Figure 4 also shows the switching circuit, which can be implemented using a 555 integrated circuit as it is reported in [27]. Figure 4 highlights the calculation of the adaptive term $k$, which is used to adapt the switching function to changes in the duty cycle. In addition, such implementation uses the normalized parameters $\alpha$ and $\beta$ as static gains, and those are used to calculate, in real-time, the parameters $a$ and $b$. The figure puts into evidence the final calculation of the switching function as $X = i_m - i_v$, where $i_v$ is the term $-a \cdot \left( v_{bus} - v_{ref} \right) - b \cdot \int \left( v_{bus} - v_{ref} \right) dt$, and $i_m$ changes from $i_b$ to $n \cdot i_k$ depending on $u$ as discussed before. Finally, the switching circuit generates the binary control signal $u$ and the complementary signal $\bar{u}$ to control the battery interface reported in Figure 1.

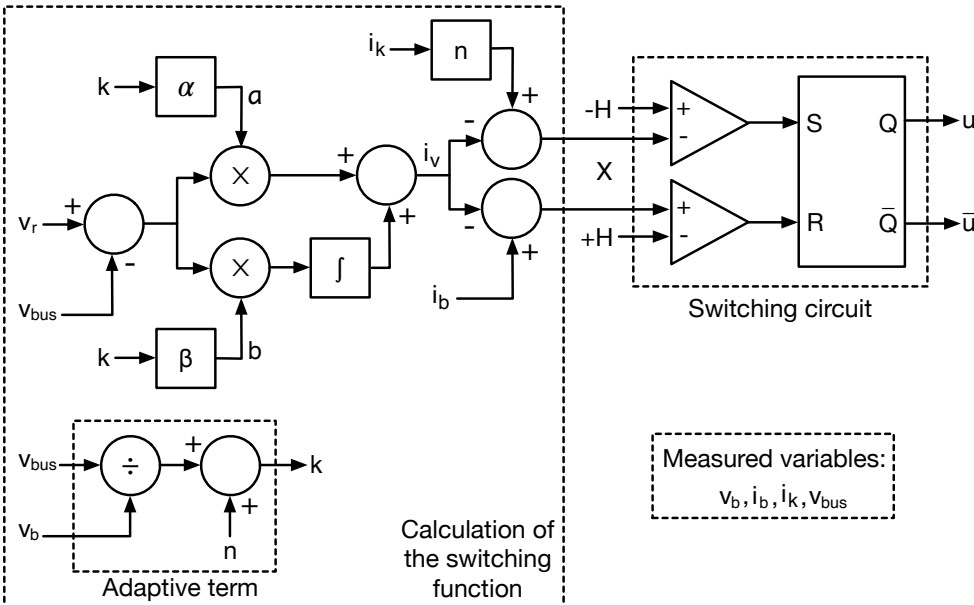

**Figure 4.** Practical implementation of the proposed SMC.

Therefore, the proposed charging/discharging system, with the controller described in Section 4, the design procedure described in Section 5, and the implementation introduced in this Section, has advantages over other flyback-based charging (C), discharging (D), or charging/discharging (C/D) systems reported in the literature. Table 1 shows the main characteristics of those systems: type (i.e., C/D, C, or D), the controller and the variable(s) regulated by it, if the controller includes the design procedure or not, the measured voltages and currents concerning the flyback converter, if the controller guarantees global stability for any operating condition or not, if the controller guarantees a desired $t_s$ and maximum overshoot in the flyback output voltage ($\Delta V_{out}$) or not, the main controller limitation, and the reference (column Ref. of the table).

Regarding the charging-discharging systems proposed in [12] and [13] (rows 2 and 3), the proposed controller can be applied for an existing converter or a new converter and it does not require the measurement of $i_{bus}$, which may be difficult to obtain since it is the algebraic sum of the currents incoming to and outgoing of the dc bus. Additionally, the proposed converter does not use a cascade controller; hence, $t_s$ is not limited by the crossover frequencies of the inner and outer loops like the controller introduced in [13]. Table 1 also shows that most of the charging and discharging systems use linear controllers (rows 4 to 11) in cascade, like in [16,22,23] (rows 7 to 9), or in a single loop, where PI is the most widely used regulator; while the charging systems proposed in [19,26] use peak current controllers (PCC). Most of those controllers do not provide a design procedure, except for [24], which uses trial and error to determine the PI parameters, and [19], which introduces a design procedure of both, the converter and controller parameters.

Analyzing the controllers reported in the charging and discharging systems shown in Table 1 (rows 4 to 13), it is possible to identify that they do not guarantee global stability for any operating condition since most of the controllers are linear and tuned for a linearized model of the system for a particular operating point. Moreover, considering that most of the controllers do not include a design procedure, and the ones that include it do not perform stability analysis, the controllers shown in rows 4 to 13 cannot guarantee the desired $t_s$ and $\Delta V_{out}$; therefore, the dynamic response of those systems cannot fulfill a desired dynamic behavior.

**Table 1.** Controllers comparison of flyback-based battery charging, discharging, and charging-discharging systems.

| Type | Controller | Design Procedure | Measured Variables | Global Stability | Desired $t_s$ | Desired $\Delta V_{out}$ | Main Limitation | Ref. |
|---|---|---|---|---|---|---|---|---|
| C/D | A. SMC of $v_{bus}$ | Yes | $v_{in}, v_{out}, i_{in}, i_{out}$ | Yes | Yes | Yes | - | TP |
| C/D | A. SMC of $v_{bus}$ | Yes: flyback and SMC | $v_{in}, v_{out}, i_{in}, i_{out}, i_{bus}$ | Yes | Yes | Yes | For new converters only | [12] |
| C/D | CC: A. P of $i_m$, A. PI of $v_{bus}$ | Yes | $v_{in}, v_{out}, i_{in}, i_{out}, i_{bus}$ | Yes | Yes | Yes | Limited $t_s$ | [13] |
| D | PI of $v_{bus}$ | No | $v_{out}$ | No: FLC | No | No | No desired DB | [15] |
| D | 2P2Z of $v_{bus}$ | No | $v_{out}$ | No: FLC | No | No | No desired DB | [17] |
| D | 2P2Z of $v_{bus}$ | No | $v_{out}$ | No: FLC | No | No | No desired DB | [18] |
| C | CS: PI of $i_b$, PI of $v_b$ | No | $v_{in}, i_{in}$ | No: FLC | No | No | No desired DB | [16] |
| C | CC: PI of $i_b$, PI of $v_b$ | No | $v_{in}, i_{in}$ | No: FLC | No | No | No desired DB | [22] |
| C | CC: PI of $i_b$, PI of $v_b$ | No | $v_{in}, i_{in}$ | No: FLC | No | No | No desired DB | [23] |
| C | PI of $v_b$ | Trial and error | $v_{in}$ | No: FLC | No | No | No desired DB | [24] |
| C | PI of $v_b$ | No | $v_{in}$ | No: FLC | No | No | No desired DB | [25] |
| C | PCC E.C.R. of $v_b$ | Yes | $v_{out}, i_{in}, i_{out}$ | No | No | No | No desired DB | [19] |
| C | PCC of $v_b$ | No | $v_{out}, i_{in}$ | No | No | No | No desired DB | [26] |

C/D: charger/discharger, C: charger, D: discharger, Ref.: reference, A.: adaptive, SMC: sliding-mode controller, CC: cascade controller, 2P2Z: two poles - two zeros compensator, PCC: peak current controller, E.C.R.: exponential compensation ramp, $v_{bus}$: dc bus voltage, $i_m$: magnetizing current, $i_b$: battery current, $v_b$: battery voltage, $v_{in}$: flyback input voltage, $v_{out}$: flyback output voltage, $i_{in}$: flyback input current, $i_{out}$: flyback output current, $i_{bus}$: dc bus current, FLC: fixed linear controller, TP: this paper, DB: dynamic behavior.

## 7. Design Example and Simulation Results

This section illustrates the design of the proposed SMC using an application example. The main step to outline the example is to define the performance criteria for the dc bus, which must be in agreement with real application cases: for example, in [31] it is analyzed the performance of a dc bus used in electric ships to support pulsating loads (electronic devices, lights, among others), where a satisfactory operation of the loads is achieved with a maximum voltage deviation of 5.2%. A similar study was reported in [32], where different approaches to power flow control on electric ships are contrasted. In that case, satisfactory performance of the bus voltage is achieved for a maximum voltage deviation of 5.3% (in p.u.) with a restitution time (settling time) close to 100 ms; since the power system analyzed in that work uses a dc bus of 5 kV, such a large settling time is needed to charge the large capacitors of the bus. Finally, the work presented in [33] reports the performance of a low voltage dc bus (30 V), which is used in a shipboard dc power distribution for electronic devices, where a satisfactory performance is achieved for voltage deviations close to 4% and a settling time close to 1.5 ms.

Based on the previous literature review, the performance criteria for the example are defined as a weighted average depending on the voltage level; the proposed example adopts a classical battery voltage $v_b = 12$ V and a common bus voltage $v_{ref} = 48$ V, hence, the maximum voltage deviation is defined as $\Delta V_{bus} = 5\%$ and the settling time as $t_s$ ($\epsilon = 2\%$) = 1 ms for a perturbation of $I_{bus} = 1$ A. Finally, in agreement with such a voltage level, the parameters of the battery interface are $C_{bus} = 50$ μF, $n = 5.4$, $L_m = 20$ μH, and $L_k = 4$ μH; the maximum switching frequency $F_{sw} = 200$ kHz is selected based on the characteristic of commercial MOSFET such as the RFP2N12. In any case, it must be

highlighted that the proposed control method applies to any other criteria and parameters values. The validation of this application is based on detailed circuital simulations carried out in the professional power electronics simulator PSIM [34], which is widely used in the industry.

The summary of the design process of the SMC is reported as follows: the performance criteria of the SMC are first defined (maximum $F_{sw}$, $I_{bus}$, $\Delta V_{bus}$, and $t_s$); then the equivalent poles $\sigma_1$ and $\sigma_2$ are calculated. Using those poles' values, the normalized parameters $\alpha$ and $\beta$ are calculated, which are used in the practical implementation as reported in Figure 4. Finally, the parameters $a$ and $b$ are calculated at the nominal operating conditions, to evaluate the stability restrictions of the SMC. If any of the stability restrictions is not fulfilled, the performance criteria must be relaxed and the equivalent poles must be calculated again; instead, if all the stability restrictions are fulfilled, the SMC is globally stable and the hysteresis band is calculated to ensure a safe switching frequency. To illustrate the stability restrictions of the SMC, Figure 5 presents the stability zone of the proposed SMC, which shows the limits imposed by the reachability and transversality conditions. Moreover, the figure is restricted to the $\alpha$ and $\beta$ values ensuring the performance criteria are fulfilled (green zone), thus $\Delta V_{bus} \leq 5\%$ and $t_s \leq 1$ ms. The figure also defines a zone to be analyzed more in detail, which is near to the frontier in which $\Delta V_{bus} = 5\%$ and $t_s = 1$ ms: such a zone is selected since it is close to the values used in this example.

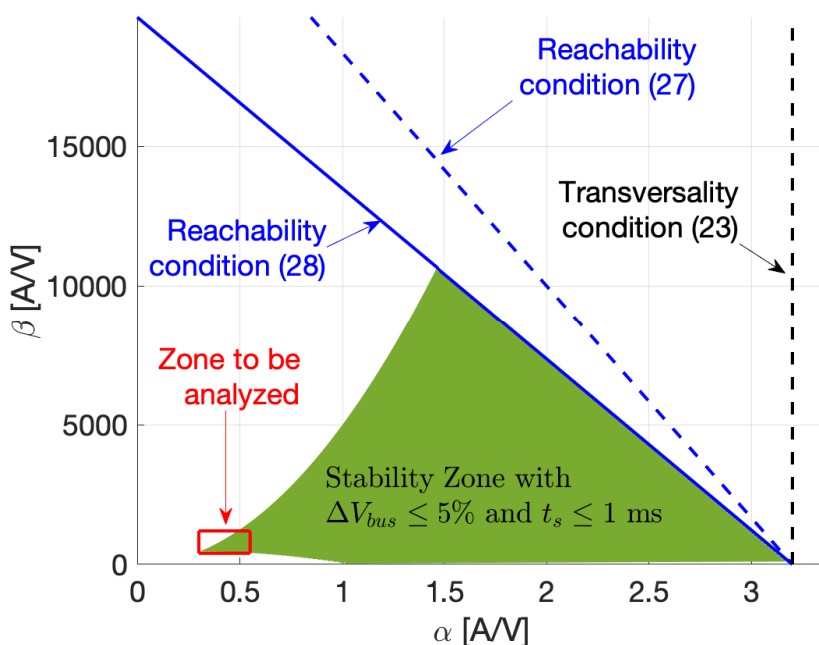

**Figure 5.** Analysis of the zone imposed by the stability restrictions for $\Delta V_{bus} \leq 5\%$ and $t_s \leq 1$ ms.

The theoretical response of the bus voltage to a 1 A step current, considering the selected $\alpha = 0.34$ A/V and $\beta = 500$ A/V parameters, was performed using the normalized transfer function (34). Figure 6 shows such a simulation, where the predicted $\Delta V_{bus} = 4.62\%$ and $t_s = 0.94$ ms performance criteria are achieved, which validates the design process for the SMC parameters proposed in Section 5. In addition, the simulation of Figure 6 also confirms that no voltage oscillations occur in the bus voltage, thus the over-damped waveform predicted in Figure 3 is also achieved. In conclusion, the simulation of Figure 6 confirms the correctness of the proposed process to calculate the SMC parameters. The final step in the design process is to calculate the hysteresis width, which results in $H = 0.65$ A. Then, the values of $\alpha$, $\beta$, and $H$ calculated in this section are used to parameterize the control scheme of Figure 4, which controls the battery interface of Figure 1. Those power and

control circuits are implemented in the power electronics simulator PSIM, which provides realistic circuital simulations of the proposed SMC.

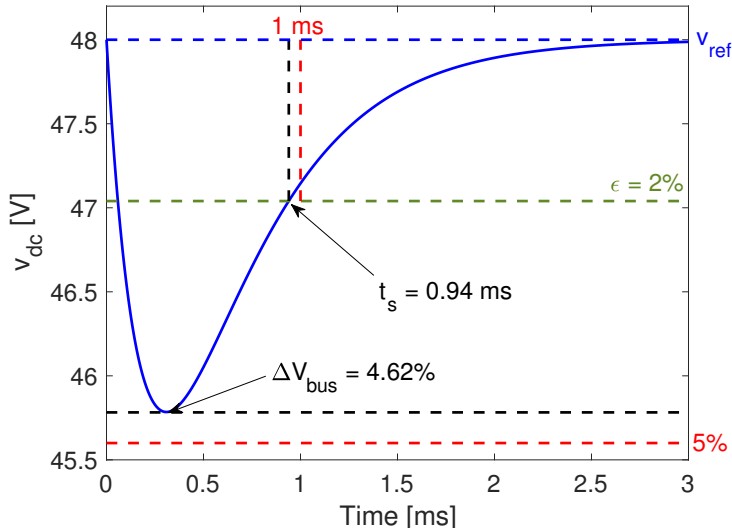

**Figure 6.** Theoretical response of the bus voltage to 1 A step with $\alpha = 0.34$ A/V and $\beta = 500$ A/V.

Figure 7 presents the first circuital simulation performed in PSIM, which considers the battery interface operating under steady-state conditions. Such a simulation verifies three aspects: (i) The switching function $X$ is correctly formed by $i_m - i_v$, with $i_m = i_b$ when $u = 1$ and $i_m = n \cdot i_k$ when $u = 0$; (ii) The control law reported in (44), since $X >= +H$ forces the control signal to change to $u = 0$, and $X <= -H$ forces the control signal to change to $u = 1$ and (iii) The switching function is always trapped inside the hysteresis band as reported in (43), thus the SMC is globally stable. In conclusion, this circuital simulation confirms the correct operation of the SMC practical implementation proposed in Section 6.

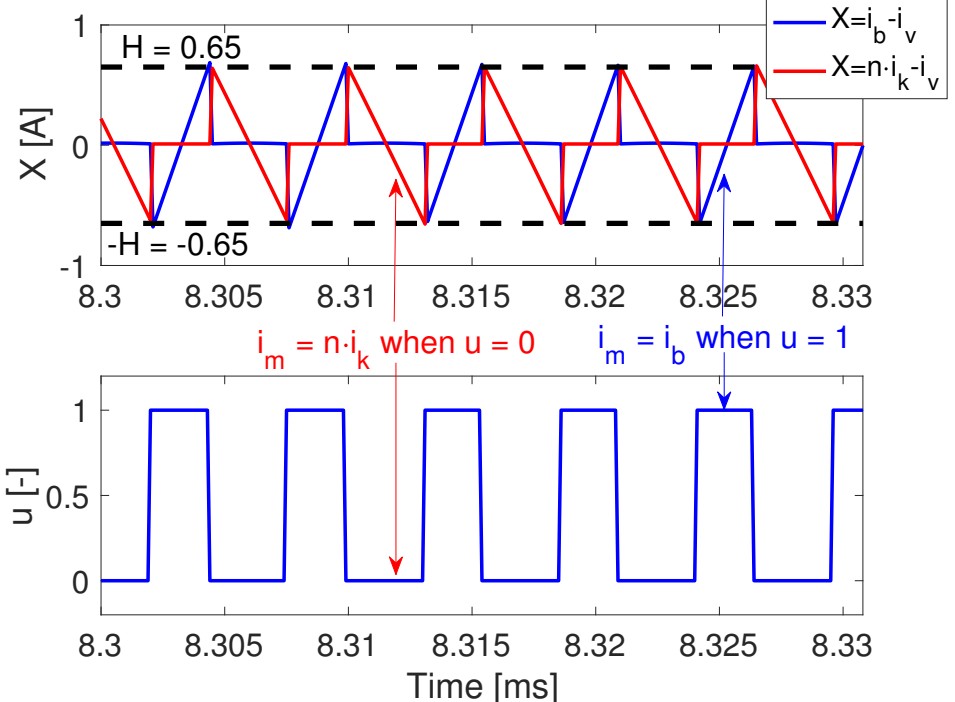

**Figure 7.** Control law verification.

A second circuital simulation, reported in Figure 8, evaluates the system response to a step current perturbation, which is the strongest possible perturbation that can be caused by any load or source. In this simulation, the step current has the amplitude $I_{bus} = 1$ A, where the circuital simulation reports the same behavior predicted by the theoretical simulation of Figure 6 for $\alpha = 0.34$ A/V and $\beta = 500$ A/V: a maximum voltage deviation $\Delta V_{bus} = 4.62\%$ and a settling time $t_s = 0.94$ ms. Therefore, the circuital simulation of Figure 8 verifies the proposed design procedure of the SMC, which enables to ensure a safe operating voltage to any device connected to the dc bus. Finally, this simulation also shows the change of the adaptive term $k$, defined in (33), which compensates for the changes on the duty cycle $d$, thus ensuring the same performance in any operating condition, i.e., for any $d$ value.

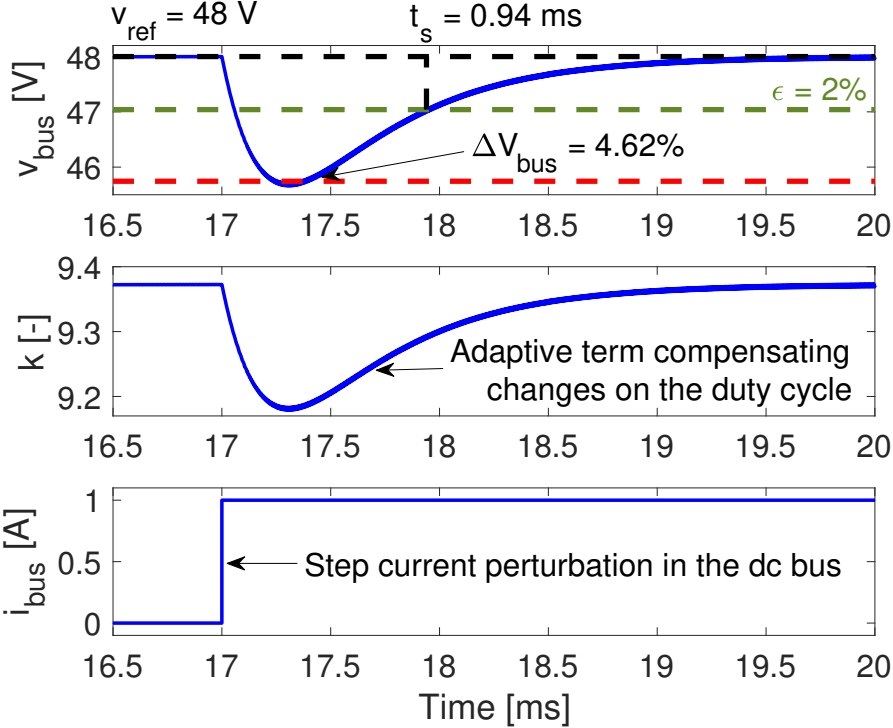

**Figure 8.** Response to step current perturbation.

A third circuital simulation was conducted to verify the correct operation of the SMC under the three possible states, i.e., battery charge, idle, and discharge. The results of such a complex simulation are reported in Figure 9, where the bus current changes from discharge condition ($i_{bus} = I_{bus} = 1$ A) to idle ($i_{bus} = 0$ A), then changes from idle to charge condition ($i_{bus} = -I_{bus} = -1$ A), changing again to both idle and discharge states. Therefore, this third simulation evaluates the SMC operation in three possible states, and the changes among those states are caused by step currents similar to the one tested in the second simulation of Figure 8. In fact, the simulation results reported in Figure 9 shows that the maximum deviation and settling time of the bus voltage are $\Delta V_{bus} = 4.62\%$ and $t_s = 0.94$ ms independent of the initial and final operation state, which confirms that the SMC provides the same bus voltage performance for any operating condition. Therefore, the proposed SMC could ensure a safe operating condition for all the devices connected to the bus, independent of the power flow exchanged with the battery. Finally, the simulation confirms the correct design of the hysteresis band to limit the switching frequency to the value defined above. In addition, the simulation also confirms that $H$ must be designed in charge condition ($i_{bus} < 0$) since in such a state occurs the highest switching frequency, which confirms the correctness of the design equation for $H$ given in (46). Finally, the theoretical and circuital simulations presented in this section confirm the stability of the SMC. Similarly, the design process provides a simple procedure to calculate the SMC parameters needed to ensure the desired performance of the bus voltage, which could be defined depending

on the characteristics of the devices connected to the bus, thus providing safe conditions under any operation state.

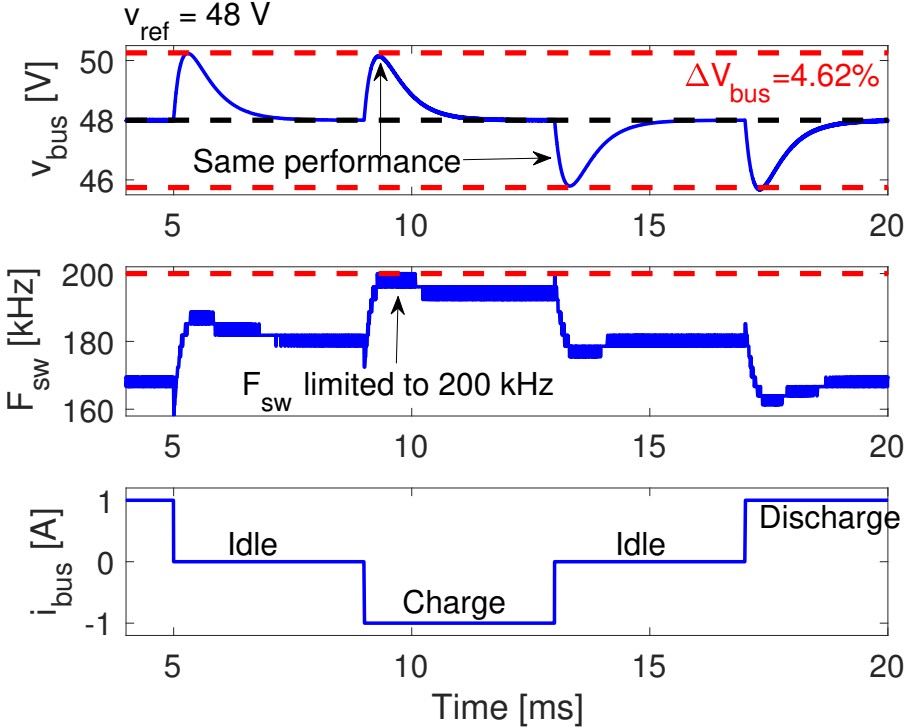

**Figure 9.** Operation under charge, idle, and discharge conditions.

## 8. Conclusions

This paper presented an adaptive SMC for a flyback-based dc bus voltage regulation system as well as the design process to calculate the SMC parameters, which can be applied to a flyback converter already designed. In this way, the proposed solution can be used in different applications where dc bus voltage regulation is required without changing the power interface. The proposed approach considers the magnetizing current as part of the switching function which provides a precise description of the converter operation.

The results obtained from detailed circuital simulations validate the accuracy in the design process for the SMC parameters since the predicted $\Delta V_{bus}$ and $t_s$ performance criteria are achieved. In the same way, those simulations allowed confirming that no voltage oscillations occur in the bus voltage, which improves the performance of the converter. Also, the simulation results confirmed the stability of the SMC, which in turn validated the design criteria proposed in this work. In this way, the proposed design procedure of the SMC enables ensuring a safe operating voltage to any device connected to the dc bus. On the other hand, the analysis of the performance of the SMC showed how the adaptive term $k$ compensates for the changes on the duty cycle $d$, thus ensuring the same performance in any operating condition. Then, the SMC could ensure a safe operating condition for all the devices connected to the bus, independent of the power flow exchanged with the battery. In addition, results confirmed that $H$ must be designed in charge condition ($i_{bus} < 0$), which corresponds to the highest switching frequency state. Finally, additional verifications based on experimental prototypes will be performed in the future to evaluate the performance of the proposed solution operating in commercial hardware.

The proposed process allows the calculation of the SMC parameters in a simple way ensuring the desired performance of the bus voltage. In this way, considering the operating characteristics of the devices connected to the bus, it is possible to guarantee safe conditions under any operation state.

**Author Contributions:** Conceptualization, C.A.R.-P., J.D.B.-R. and L.A.T.-G.; methodology, C.A.R.-P. and J.D.B.-R.; validation, C.A.R.-P. and J.D.B.-R.; writing—original draft preparation, C.A.R.-P., J.D.B.-R. and L.A.T.-G.; writing—review and editing, C.A.R.-P., J.D.B.-R. and L.A.T.-G. All authors have read and agreed to the published version of the manuscript.

**Funding:** This work was supported by the Universidad Nacional de Colombia and Minciencias (Fondo nacional de financiamiento para ciencia, la tecnología y la innovación Francisco José de Caldas) under the projects "Estrategia de transformación del sector energético Colombiano en el horizonte de 2030 - Energetica 2030"—"Generación distribuida de energía eléctrica en Colombia a partir de energía solar y eólica" (Code: 58838, Hermes: 38945).

**Institutional Review Board Statement:** Not applicable.

**Informed Consent Statement:** Not applicable.

**Data Availability Statement:** The data used in this study are reported in the paper figures and tables.

**Acknowledgments:** Authors thank to the Universidad Nacional de Colombia and the Instituto Tecnológico Metropolitano.

**Conflicts of Interest:** The authors declare no conflict of interest.

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
