# Peer review of "Sliding-Mode Control of Bidirectional Flyback Converters with Bus Voltage Regulation for Battery Interface"

_computation, doi:10.3390/computation10070125_

Round 1
Reviewer 1 Report
1, It is better to change the battery bank to the battery pack.
2, In figure 9, what are the criteria for maximum derivation and setting time?
3, Can you summarize the performances of other methods in published journal papers? It is better to summarize them and make a comparison. If not, it is difficult to identify the originality of this manuscript
Reviewer 2 Report
Missing experimental validations.
Put some preliminary experimental results.
Make some comparison between simulation and experiments.
Missing comparison between your results and literature results.
Round 2
Reviewer 2 Report
The experiments validate the theoretical hypotheses. I understand that you cannot perform experimental validation.